# Prenatal and Postpartum Maternal Iodide Intake from Diet and Supplements, Urinary Iodine and Thyroid Hormone Concentrations in a Region of the United Kingdom with Mild-to-Moderate Iodine Deficiency

**DOI:** 10.3390/nu13010230

**Published:** 2021-01-14

**Authors:** Diane E. Threapleton, Dagmar Waiblinger, Charles J.P. Snart, Elizabeth Taylor, Claire Keeble, Samina Ashraf, Shazia Bi, Ramzi Ajjan, Rafaq Azad, Neil Hancock, Dan Mason, Stephen Reid, Kirsten J. Cromie, Nisreen A. Alwan, Michael Zimmermann, Paul M. Stewart, Nigel A.B. Simpson, John Wright, Janet E. Cade, Laura J. Hardie, Darren C. Greenwood

**Affiliations:** 1Leeds Institute of Cardiovascular & Metabolic Medicine, School of Medicine, University of Leeds, Leeds LS2 9JT, UK; D.E.Threapleton@leeds.ac.uk (D.E.T.); charles.snart@gmail.com (C.J.P.S.); lizzietaylor_1@msn.com (E.T.); C.M.Owen@leeds.ac.uk (C.K.); R.Ajjan@leeds.ac.uk (R.A.); K.J.Cromie@leeds.ac.uk (K.J.C.); L.J.Hardie@leeds.ac.uk (L.J.H.); 2Bradford Institute for Health Research, Bradford Teaching Hospitals NHS Foundation Trust, Bradford BD9 6RJ, UK; Dagmar.Waiblinger@bthft.nhs.uk (D.W.); Samina.Ashraf@bthft.nhs.uk (S.A.); Shazia.Bi@bthft.nhs.uk (S.B.); Rafaq.Azad@bthft.nhs.uk (R.A.); Dan.Mason@bthft.nhs.uk (D.M.); John.Wright@bthft.nhs.uk (J.W.); 3Leeds Institute for Data Analytics, University of Leeds, Leeds LS2 9JT, UK; 4Nutritional Epidemiology Group, School of Food Science & Nutrition, University of Leeds, Leeds LS2 9JT, UK; N.Hancock@leeds.ac.uk (N.H.); J.E.Cade@leeds.ac.uk (J.E.C.); 5Earth Surface Science Institute, School of Earth and Environment, University of Leeds, Leeds LS2 9JT, UK; S.Reid@leeds.ac.uk; 6Faculty of Medicine, University of Southampton, Southampton SO16 6YD, UK; N.A.Alwan@soton.ac.uk; 7NIHR Southampton Biomedical Research Centre, University of Southampton and University Hospital Southampton NHS Foundation Trust, Southampton SO16 6YD, UK; 8Laboratory for Human Nutrition, Institute of Food, Nutrition and Health, ETH Zurich, 8092 Zurich, Switzerland; michael.zimmermann@hest.ethz.ch; 9Faculty of Medicine and Health, University of Leeds, Leeds LS2 9JT, UK; P.M.Stewart@leeds.ac.uk; 10Division of Women’s and Children’s Health, School of Medicine, University of Leeds, Leeds LS2 9JT, UK; N.A.B.Simpson@leeds.ac.uk

**Keywords:** iodine, pregnancy, diet, thyroid, cohort

## Abstract

Iodine is essential for normal thyroid function, supporting healthy fetal and child development. Iodine requirements increase in pregnancy, but many women in regions without salt iodization have insufficient intakes. We explored associations between iodide intake and urinary iodine concentration (UIC), urinary iodine/creatinine ratio (I/Cr), thyroid stimulating hormone, thyroglobulin, free triiodothyronine, free thyroxine and palpable goiter in a region of mild-to-moderate iodine insufficiency. A total of 246 pregnant women aged 18–40 in Bradford, UK, joined the Health and Iodine in Babies (Hiba) study. They provided detailed information on diet and supplement use, urine and serum samples and were assessed for goiter at around 12, 26 and 36 weeks’ gestation, and 6, 18 and 30 weeks postpartum. Dietary iodide intake from food and drink was estimated using six 24 h recalls. During pregnancy, median (IQR) dietary iodide intake was 101 µg/day (54, 142), with 42% from dairy and 9% from white fish. Including supplements, intake was 143 µg/day (94, 196), with 49% < UK reference nutrient intake (140 µg/day). Women with Pakistani heritage had 129 µg/day (87, 190) median total intake. Total intake during pregnancy was associated with 4% (95% CI: 1%, 7%) higher UIC, 5% (3%, 7%) higher I/Cr, 4% (2%, 6%) lower thyroglobulin and 21% (9%, 32%) lower odds of palpable goiter per 50 µg/day. This cohort consumed less iodide in pregnancy than UK and World Health Organization dietary recommendations. UIC, I/Cr and thyroglobulin were associated with intake. Higher intake was associated with fewer goiters. Because dairy was the dominant source of iodide, women following plant-based or low-dairy diets may be at particular risk of iodine insufficiency.

## 1. Introduction

Iodine is essential for normal thyroid function, thyroid-mediated growth and metabolism at all stages of life [1]. Adequate iodide intake in pregnancy and lactation is especially important to meet heightened demands caused by increased renal clearance during pregnancy and to support growth and development in the fetus [2]. Severe iodine deficiency has long been associated with detrimental and irreversible consequences for offspring, including hypothyroidism and damage to the developing brain. Iodine deficiency is also associated with greater risk of stillbirth, delayed physical development and impaired mental function [3]. However, less is known about any consequences for fetal development of mild or moderate deficiency. Associations with smaller birthweight [4,5,6], preterm birth [5] and child neurodevelopment [7,8,9] have been observed in mildly or moderately deficient populations, but evidence is inconsistent, with other large studies reporting no evidence of associations [10,11,12,13].

The World Health Organization (WHO) recommends that pregnant and lactating women should have an iodide intake of 250 µg/d and defines populations with adequate intake as having a median urinary iodine concentration (UIC) greater than 150 µg/L [3], largely based on avoidance of goiter. However, there is limited evidence describing how these criteria are reflected at a functional level in terms of thyroid hormones, or how iodine requirements vary during pregnancy and lactation [1]. Because of the limited evidence, the UK does not currently set pregnancy-specific dietary guidelines for iodide intake, but instead the UK reference nutrient intake (RNI) of 140 µg/d for all adults is used, with a lower RNI (LRNI) of 70 µg/day [1,14]. The National Diet and Nutrition Survey (NDNS) indicates that 15% of adult women in the UK fall short of the LRNI [15], with a greater proportion of Black (23%) or Asian (18%) women below the LRNI than White women (9%) [16]. A recent systematic review concluded that, despite methodological differences between studies, pregnant women in the UK are generally iodine insufficient [17]. The UK is not alone in this, with Norway in a similar position [18,19]. This may reflect the lack of any fortification or salt-iodization program in both the UK and Norway, in contrast to many other developed countries. The range of foods rich in iodide is limited to mostly seafood, dairy and eggs [20], so intakes may be particularly low amongst those following restricted or plant-based diets [21,22]. Without larger-scale monitoring programs, it is challenging to quantify iodine status in vulnerable sub-groups. Since iodine demand increases in pregnancy, there is more general concern about the iodine status of UK women of reproductive age [1]. Whilst some recent studies report urinary iodine status in pregnant populations [11,23,24], few have examined how iodide intake and urinary iodine status vary across different trimesters and postpartum and how these relate to thyroid function.

We aimed to address these key uncertainties by characterizing current iodine status throughout pregnancy and postpartum, comparing women of different ethnic groups, assessing how this differs by supplement use and dietary preferences, and examining how urinary iodine status, thyroid hormones and protein concentrations relate to iodide intake.

## 2. Materials and Methods

### 2.1. Participants

The Health and Iodine Status of Babies (Hiba) longitudinal study recruited pregnant women in Bradford between August 2016 and October 2017 and captured seasonal variation in diet and iodine status. Women were approached at their dating ultrasound appointment (typically 12 weeks’ gestation), and participants remained in the study until 30 weeks postpartum. Recruitment targeted women with a confirmed pregnancy, between 9 and 15 weeks’ gestation, aged 18–40 years and with no previous diagnosis or first-degree relative history of a thyroid condition. All participants provided informed consent, and the study was approved by the Yorkshire and the Humber Research Ethics Committee (16/YH/0260). The study protocol was registered and published online (NCT registration: 03552341).

Participants were visited at home or in the clinic by a qualified research nurse six times during the course of the study, within 3 weeks either side of the target date at 12, 26 and 36 weeks’ gestation and 6, 18 and 30 weeks postpartum. At the first visit, detailed lifestyle, demographic and anthropometric data were collected. At each visit, the nurse used standard methods to administer a repeat questionnaire, conduct a 24 h diet recall and collect non-fasting blood and spot urine samples for analysis of urinary iodine status and thyroid measures (details below). Nurses that spoke Urdu and/or Mirpuri visited women that did not speak English, and translated versions of questionnaires were used.

### 2.2. Dietary Assessment

For each study visit, the research nurse administered a multiple-pass 24 h recall face-to-face to record all food and drink consumed by the participant in the previous day. The dietary information was recorded on the myfood24 tool, to allow for efficient conversion of nutrient outputs [25]. The tool has previously been validated against a range of biomarkers and traditional 24 h recalls, with good agreement and comparable nutrient estimates, including iodide [26,27]. Myfood24 is supported by a database of over 50,000 products available in UK shops, utilizes manufacturer-specific composition and portion sizes and can automatically calculate nutrient intakes. Nutrients were calculated based on UK food composition tables (Composition of Foods Integrated Dataset (CoFID) v.7) [20].

Where foods in CoFID v.7 and branded products lacked estimates of iodide content, the nearest similar items in the database were used. For instance, fat-free fruit yogurt and battered haddock do not have iodine values in CoFID v.7, so the iodide content of fat-free plain yogurt and bread-crumbed haddock were used instead. All foods consumed were then categorized into 10 broad sources of iodide including dairy foods, white fish, eggs and egg dishes, with composite foods given their own categories (food category descriptors are detailed in Appendix A). Dietary patterns such as vegan and vegetarian were self-reported at visit 1.

### 2.3. Iodide Intake from Supplements

Participants provided details of the type, brand, dose and frequency of all multivitamin/mineral supplement use at each study visit. The iodide content of all reported supplements was identified from manufacturer and retailer websites. Supplements were categorized as those containing iodide and those without any. The amount of iodide was quantified based on published ingredients.

### 2.4. Measurement of Urinary Iodine

Frozen urine aliquots were transferred from storage in Bradford to the Leeds Institute of Cardiovascular and Metabolic Medicine, University of Leeds, UK, for iodine and creatinine analysis. While urinary iodine concentration (UIC) is commonly used to identify populations with insufficient iodide intakes, iodine-to-creatinine ratio (I/Cr) has been found to more correctly reflect 24 h urine excretion in pregnant women [28], and so both measures were used. Urinary iodine-127 concentration (µg/L) was calculated using inductively coupled plasma mass spectrometry (ICP-MS) (Thermo iCAP Q, Hemel Hempstead, UK). The instrument method was accredited by the Centers for Disease Control and Prevention (CDC) EQUIP international standardization program, achieving a 100% success rate for 2017/2018. A standard Jaffe reaction-based microplate assay was used to evaluate urinary creatinine concentrations. Detailed methods for urinary iodine and creatinine measurement have been described previously [11].

### 2.5. Thyroid Measures

Concentrations of thyroid stimulating hormone (TSH), thyroglobulin (Tg), free triiodothyronine (fT3) and free thyroxine (fT4) were determined in serum samples through pre-validated commercial ”sandwich” enzyme linked immunosorbent assays (ELISAs) according to the manufacturers’ instructions (Biovendor, Czech Republic, Cat Nos: RCD028R, RCD014R, RCD015R and AbCam, UK, Cat No: Ab155441). The presence of goiter was assessed at each study visit using a standardized clinical procedure that involved inspection and palpation of the thyroid gland, according to local endocrine practice guidelines. Training and assessment of the research nurses was provided by a qualified endocrinologist. Goiters were graded according to WHO criteria [29,30].

Certified reference materials for fT3, fT4 and TSH ELISAs consisted of Seronorm Immunoassay Lyo L-1, L-2 and L-3 and a separate CRM, Thyroglobulin BCR457, for thyroglobulin ELISAs. Intra-assay coefficients of variation (CVs) at three separate QC concentrations were 9.7%, 7.0% and 3.5% for fT3; 4.8%, 4.9% and 4.3% for fT4; 13.3%, 6.4% and 7.7% for TSH and 2.8%, 4.5% and 2.4% for Tg. Inter-assay coefficients of variation were 8.6%, 7.8% and 8.2% for fT3; 12.3%, 11.5% and 9.9% for fT4; 8.3%, 12.3% and 12.8% for TSH and 13.5%, 11.5% and 19.9% for Tg.

All available samples from each participant were analyzed together to avoid inter-run variation.

### 2.6. Statistical Methods

Average intakes are presented as geometric means (GEMs) with 95% confidence intervals (CIs) and medians with interquartile ranges (IQRs) to facilitate comparison with WHO definitions. Where zero intake was possible, a constant 1 was added to all values before computing the geometric mean. Estimates were explored by demographic characteristics, by time (study visits before and after delivery) and by ethnic group.

Associations between iodide intake and urinary iodine measures (UIC and I/Cr) or thyroid hormone and protein concentrations (TSH, Tg, fT3, fT4 and fT3/fT4 ratio) were explored using linear mixed-effects (multilevel) models to account for repeated measures within individuals, with random slopes for each exposure. Model covariates were based on known confounders and likely competing exposures in the association between iodide intake and these urinary or blood measures. Adjustments included maternal age, body mass index (BMI) at first visit, parity, ethnic background, highest educational achievement and National Statistics Socio-economic Classification (NS-SEC). Models of continuous outcomes also included smoking status and alcohol use (trimester 1 or after birth, depending on model). Models using postnatal data were additionally adjusted for gestation length. To adjust for seasonality, pairs of sine and cosine functions were prepared for each date across the year [31]. Urinary iodine or thyroid hormone concentrations were transformed using the natural logarithm and estimates subsequently presented as percent differences (e.g., percent difference in I/Cr per 50 µg increase in total iodide intake). An increment of 50 µg/day iodide intake was selected as a round number approximately equivalent to the difference between the first and second quartiles of intake, allowing meaningful presentation of parameters from regression models. Continuous covariates were centered on the mean, and categorical covariates used the largest group as the reference.

Sensitivity analyses were conducted excluding twin pregnancies and including adjustment for total energy intake. All data preparation and analyses were conducted using Stata IC version 15.1 (Stata Corp, College Station, TX, USA).

### 2.7. Sample Size Requirements

Required sample size estimates were based on previously published data of between-individual and within-individual variances for UIC from spot urine [32]. Three measurements during pregnancy from 200 participants and three after delivery allow estimation of the geometric mean UIC to within ±4%, and estimation of mean iodine status at a single timepoint ±6%, allowing good precision to investigate changes during and after pregnancy. A total of 200 participants also provides approximately 80% power to detect a 15% difference in UIC (equating to approximately 20 µg/L for an average woman), representing good power to detect small differences. With half of the participants having Pakistani heritage, this provided estimated geometric mean UIC for ethnic subgroups to within approximately ±7%. To allow for anticipated attrition of around 30% during the study, the initial recruitment target was 300 women.

## 3. Results

### 3.1. Participant Characteristics at Recruitment

In total, 246 mothers were recruited, with 206 (84%) retained to the third visit at 36 weeks’ gestation and 192 (78%) retained to the final visit, 30 weeks postpartum (Appendix A). At recruitment, participants had a mean age of 31 and BMI of 27 kg/m^2^ (Table 1). For 79 (32%) this was their first pregnancy, 131 (53%) had Pakistani heritage, 89 (36%) were White British or European and 26 (11%) were from other ethnic backgrounds. Of those recruited, 105 (43%) were educated beyond the age of 18; 81 (33%) had managerial, administrative or professional occupations and 83 (34%) had never worked. At recruitment in trimester 1, 11 (4%) followed vegetarian or vegan diets, 15 (6%) smoked and 17 (7%) consumed alcohol (Table 1). There was a wide range of iodide intake at recruitment, with women in the lowest third of total iodine intake consuming 43 µg/day (95% CI 37 to 51) compared to 253 µg/day (233 to 276) for those in the highest third. There were no substantive differences in demographic characteristics between women with lower intakes and those with higher intakes, but women with lower intake had slightly higher BMI.

### 3.2. Dietary Intake During Pregnancy

The geometric mean (95% CI) total iodide intake during pregnancy was 136 µg/day (126 to 146), with a median (IQR) of 143 µg/day (94 to 196) (Table 2). Almost all (219, 89%) of the participants consumed less iodide from all sources than the WHO recommendation for pregnant and lactating women (250 µg/day) [3] (Appendix A). The UK adult RNI (140 µg/day) was not met by 119 (48%) women, with 28 (11%) below the UK LRNI (70 µg/day). Participants who did not use any iodide supplements were twice as likely (77% vs. 33%) to be below the UK RNI and three times as likely (22% vs. 6%) to be below the LRNI.

Of the total iodide intake during pregnancy, a geometric mean 96 µg/day (89 to 105) and median 101 µg/day (64 to 142) was from dietary sources (Table 2). Iodide intake remained broadly stable throughout pregnancy (Figure 1). The main dietary sources of iodide during pregnancy were dairy (42% of total intake), eggs (12%), dairy with cereal and/or egg (11%), poultry and red meat (10%) and white fish (9%) (Table 2). These sources also remained stable throughout pregnancy. None of the women used iodized salt.

### 3.3. Supplement Use

In terms of supplements, 80 (33%) used some form of vitamin or mineral supplement prior to their pregnancy (with or without iodide in it), with 26 (11%) of these using a supplement containing iodide. Supplement use was higher during pregnancy (Table 2), with the proportion of women using iodide-containing supplements highest in trimester 1, halving by the end of trimester 3, and decreasing further in the postnatal period (Table 2). For those consuming iodide-containing supplements, these contributed approximately half of their total iodide intake in the first trimester (Figure 1 and Figure 2). Women consuming supplements were typically older, had lower BMI, were more likely to have White British or European heritage, had higher educational qualifications with more managerial, administrative or professional careers and were more likely to have consumed supplements in the 3 months prior to their pregnancy. Iodide intake from dietary sources was similar between supplement users and non-users (Appendix A).

### 3.4. Subgroup Analysis

Total iodide intake was lower in women with Pakistani heritage (Table 3), the proportion with total intake below the UK RNI was also greater, intake from dietary sources was lower and slightly fewer women used supplements containing iodide. Women of Pakistani heritage obtained iodide less from dairy foods and more from eggs (Appendix A) than other ethnic groups. Iodide intake from white fish was lowest among White British and European women and higher among women in other ethnic groups.

### 3.5. Thyroid Hormones, Proteins and Goiter

The median UIC across all stages of pregnancy fell below the WHO threshold of 150 µg/L for identifying populations with adequate iodine nutrition in pregnancy [3]. Total, dietary and supplemental iodide intakes were associated with both UIC and I/Cr biomarkers of iodine status (Table 4), with 4% (1 to 7) higher UIC and 5% (3 to 7) higher I/Cr per 50 µg/day total iodide intake during pregnancy, with similar-sized associations for both dietary and supplemental sources. Of the thyroid hormones and proteins, total iodide intake was most strongly associated with thyroglobulin, with 4% (2% to 6%) lower thyroglobulin per 50 µg/day iodide intake. Across all three trimesters, 89 (36%) women were identified as having a palpable goiter (Appendix A). There was some evidence of an association between total iodide intake during pregnancy, particularly dietary iodide, and lower odds of palpable goiter during pregnancy, with 21% (9 to 32) lower odds per 50 µg/day (Table 4).

## 4. Discussion

In this longitudinal cohort, providing contemporary data on iodide intake and dietary habits of pregnant women in the UK, participants consumed less iodide than UK and World Health Organization dietary recommendations. Whilst women taking iodide supplements during pregnancy had substantially higher iodide intakes than those who were not, total intake was still low for many women. The Hiba cohort is not only unique in that it provides contemporary data on iodide intake and dietary habits of pregnant women in the UK, but it is also a multi-ethnic cohort, having good numbers of both women with White British and Pakistani heritage. The study therefore builds upon the NDNS findings of differences in the proportion of women from different ethnic backgrounds below the LRNI [16]. Dietary and supplemental intake was lowest in women with Pakistani heritage.

Dairy and dairy products were the main source of dietary iodide. Though white fish and other seafood is rich in iodide, intake was typically low in our cohort, so it did not make a substantial contribution to dietary iodide. Whatever the source, higher iodide intake was associated with higher UIC and I/Cr. Lower iodide intake was also associated with higher thyroglobulin and prevalence of goiter, suggesting that the thyroid is overstimulated when iodide intake falls short of demand. The association between iodide intake and goiter was stronger with dietary sources, suggesting iodide supplements may have been introduced after goiter was identified. For logistical reasons, we did not conduct an ultrasound assessment of thyroid size, which could have provided more precise estimates of associations.

The lower than recommended total iodide intakes in this prospective longitudinal cohort of pregnant women strengthens the evidence from older cross-sectional studies in the UK [33,34]. While iodide-containing supplements increased total iodide intake substantially, total intake remained lower than recommendations, and half the supplements taken did not contain any iodide, e.g., just folic acid or iron. None of the women used salt fortified with iodide, which could have provided an additional source of intake [35].

Our cohort participants had lower dietary intakes than those observed among women aged 19–64 in the UK NDNS and a larger proportion of women with intakes below the LRNI [15]. As in the Hiba study, the UK NDNS identified “milk and milk products” were the largest contributor to total iodide intake, with a relatively low contribution from fish because of generally low intakes. All studies in pregnant UK populations are therefore consistent with ours in illustrating the potential for women with restricted diets, for example plant-based or dairy-free diets, to be at greater risk of inadequate iodide intake. Some women following plant-based or dairy-free diets may therefore struggle to meet even the LRNI without supplementation.

In terms of urinary iodine status, despite lower dietary intake, we reported higher UIC than women aged 19–64 in NDNS and similar UIC to that reported for their women of childbearing age, possibly because of the addition of iodide supplements for some women in pregnancy, while NDNS excluded pregnant and lactating women [15]. Our cohort also had higher UIC than some other studies of pregnant UK populations [21,36,37] but slightly lower than another using similar methods across three UK cities [11]. Whilst there appears to be some geographical variation, our results are consistent with other locations in the UK, and there is no evidence that results from our cohort are unrepresentative. Seasonal variation was taken into account in the modelling.

Through the strong relationships established between the mothers and the experienced research team in Bradford, participant retention was excellent, so the required power was achieved across six time-points during and after pregnancy. We were therefore able to explore associations between dietary and supplemental iodide intake with urinary iodine status and thyroid function in detail. Furthermore, our dietary assessment tool covered a wide range of branded and non-branded food items, potentially providing better estimates of iodide content and enabling a full breakdown of dietary sources contributing to total intake. In addition, our estimated supplemental intake accounted for brand, frequency and dose at each time-point.

Whilst the proportion continuing breastfeeding to the end of the study was too low to explore differences with those choosing not to breastfeed, iodine requirements are likely to be greater among breastfeeding women. We show that, whilst dietary intake was similar throughout pregnancy, supplement use dropped sharply afterwards. Breastfeeding women may therefore be at greater risk of deficiency during lactation, with the potential for babies who are exclusively breastfed not to receive adequate iodine. We focused on the iodine status of the mother, rather than iodide content of breastmilk or iodine status of the child. This could be a limitation were subsequent childhood developmental outcomes being assessed.

Our measure of urinary iodine status was based on spot urine for minimal participant burden and attrition. These were collected at multiple time-points to improve precision of estimates. Robust laboratory methods included external validation by the Centers for Disease Control (CDC) [38] and the laboratory conducting NDNS at Cambridge University. Furthermore, we report both UIC and I/Cr, which accounts for important differences in urine dilution by correcting for creatinine excretion [39]. The adjustment for creatinine typically provides a good approximation for UIC from a 24 h urine sample [40]. Furthermore, our estimated geometric mean I/Cr (128 µg/g) is in close agreement with that of total iodide intake (136 µg/day). Because mean daily urinary creatinine excretion is about 1.08 g in pregnant women [41], this suggests our estimated iodide intake successfully captured global iodide intake from all sources. This study goes some way to addressing knowledge gaps in how iodide intake changes during pregnancy and postpartum, what the major dietary sources are for pregnant women and therefore which population subgroups may be at greater risk of deficiency, i.e., women following plant-based or low-dairy diets. However, evidence relating to the functional significance of mild deficiency is mixed [4,5,6,7,8,9,10,11,12,13].

## 5. Conclusions

Given the known heightened demands on iodine reserves during pregnancy or lactation, the significant implications for women’s health and both fetal and child development, this evidence that a large proportion are not meeting LRNI is potentially of concern. Our quantification and timing of dietary and supplemental iodide intake across pregnancy and beyond provides a useful context for advice on routes to improve maternal iodine status.

## Figures and Tables

**Figure 1 nutrients-13-00230-f001:**
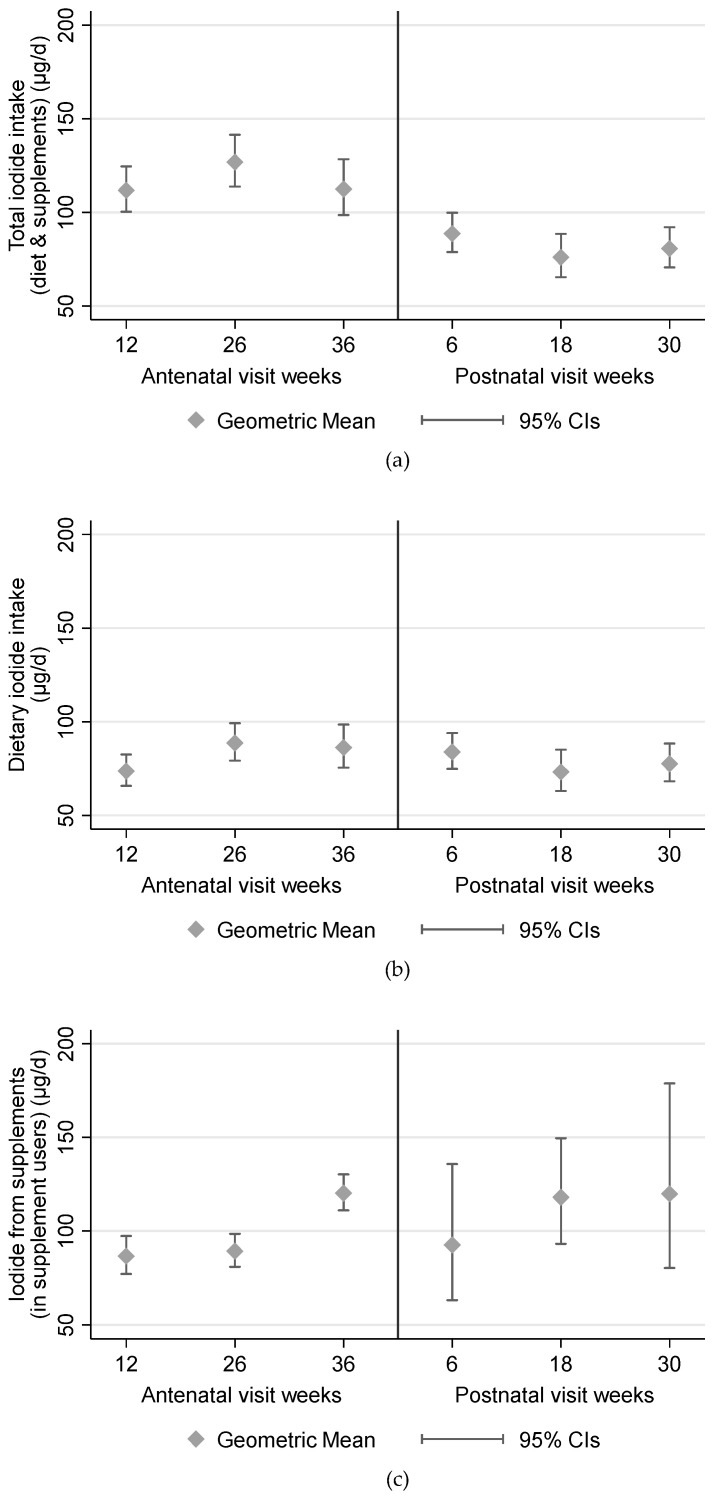
Iodide intake during pregnancy and postnatally. (**a**) Total intake; (**b**) dietary intake; (**c**) supplemental iodide intake amongst users of iodide supplements.

**Figure 2 nutrients-13-00230-f002:**
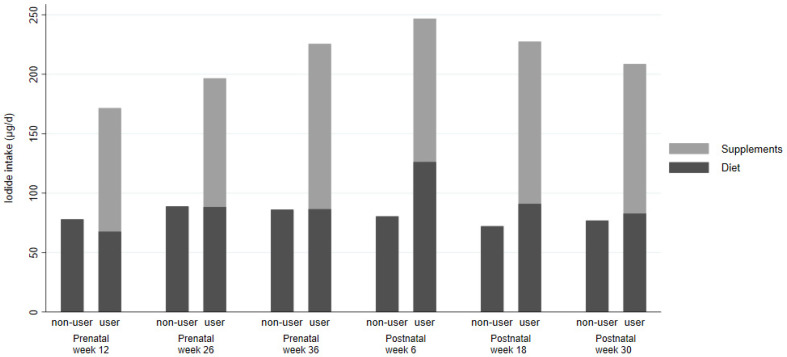
Geometric mean iodide intake from diet and diet plus supplements, at each study visit, by usage of iodine-containing supplements. Number of participants using supplements containing iodide at each study visit: *n* = 110, *n* = 98, *n* = 57, *n* = 17, *n* = 8 and *n* = 8, respectively. Number of participants not using supplements containing iodide at each study visit: *n* = 134, *n* = 119, *n* = 149, *n* = 189, *n* = 185 and *n* = 184, respectively.

**Table 1 nutrients-13-00230-t001:** Maternal characteristics by category of total iodide intake at first appointment ^1^.

		Total Iodide Intake (Diet + Supplements)		
Participant Characteristics	All	Lower Third (<91 µg/d)	Middle Third (91–162 µg/d)	Upper Third (>162 µg/d)
	(*n* = 246) ^2^	(*n* = 82)	(*n* = 81)	(*n* = 81)
Total iodide intake (µg/d) geometric mean (95% CI)	111 (100, 124)	43 (37, 51)	126 (121, 131)	253 (233, 276)
Total iodide intake (µg/d) median (IQR)	129 (73, 196)	56 (28, 73)	129 (111, 148)	239 (196, 293)
Age (years) median (IQR)	31 (27, 34)	30 (26, 34.5)	31 (28, 34)	31 (28, 34)
BMI (kg/m^2^) median (IQR)	27 (23, 31)	29 (24, 31)	27 (23, 30)	26 (23, 30)
First pregnancy, *n* (%)	79 (32)	28 (34)	21 (26)	29 (36)
Smoked in first trimester, *n* (%)	15 (6)	8 (10)	3 (4)	4 (5)
Alcohol in first trimester, *n* (%)	17 (7)	3 (4)	5 (6)	9 (11)
Vegan or vegetarian, *n* (%) ^3^	11 (4)	3 (4)	5 (6)	3 (4)
Ethnic background, *n* (%)				
White British and European	89 (36)	29 (35)	24 (30)	36 (44)
Pakistani	131 (53)	42 (51)	50 (62)	37 (46)
Other	26 (11)	11 (13)	7 (9)	8 (10)
Highest education level, *n* (%) ^4^				
<5 GCSEs or equivalent	32 (13)	12 (15)	14 (17)	6 (7)
5+ GCSEs or equivalent	58 (24)	23 (28)	17 (21)	18 (22)
A-level equivalent	43 (17)	12 (15)	15 (19)	16 (20)
Higher than A-level	105 (43)	33 (40)	30 (37)	39 (48)
Other/Don’t know	8 (3)	2 (2)	5 (6)	2 (2)
NS-SEC (*n* (%)) ^5^				
Managerial, administrative and professional	81 (33)	27 (33)	21 (26)	33 (41)
Intermediate occupations or small employers and own account workers	40 (16)	10 (12)	18 (22)	12 (15)
Lower supervisory and technical or semi-routine and routine	42 (17)	13 (16)	14 (17)	15 (19)
Never worked	83 (34)	32 (39)	28 (35)	21 (26)
TSH (µIU/mL) median (IQR)	1.15 (0.81, 1.75)	1.15 (0.87, 1.70)	1.07 (0.71, 1.50)	1.31 (0.86, 1.93)
Tg (µg/L) median (IQR)	6.21 (2.64, 12.38)	5.55 (2.79, 10.90)	8.69 (2.38, 20.88)	5.87 (2.50, 12.18)
fT3 (pg/mL) median (IQR)	2.49 (2.11, 2.87)	2.47 (2.15, 3.00)	2.49 (2.09, 2.77)	2.48 (2.07, 2.91)
fT4 (pg/mL) median (IQR)	9.69 (8.31, 11.27)	9.65 (8.23, 11.16)	9.43 (8.35, 10.96)	9.84 (8.20, 11.51)

Values are median (IQR) or *n* (%), unless otherwise stated. ^1^ Mean (SD) gestation length at first study visit was 13.9 (1.1) weeks. ^2^ Dietary intake was unavailable for *n* = 2 participants. ^3^ Does not include pescatarians. ^4^ UK-equivalized overseas qualifications. ^5^ Because of small cell counts, the National Statistics Socio-economic Classification (NS-SEC) category “intermediate occupations” was combined with “small employers and own account workers”, and “lower supervisory and technical occupations” was combined with “semi-routine and routine occupations”.

**Table 2 nutrients-13-00230-t002:** Total iodide intake, sources of iodide intake, supplemental iodide use and urinary iodine status by study visit.

	During Pregnancy				Postpartum				
	Visit 1 12 Weeks’ Gestation	Visit 2 26 Weeks’ Gestation	Visit 3 36 Weeks’ Gestation	Visits 1–3 ^1^	Visit 4 6 Weeks Postpartum	Visit 5 18 Weeks Postpartum	Visit 6 30 Weeks Postpartum	Visits 4–6 ^1^	All Visits ^1^
	(*n* = 244)	(*n* = 217)	(*n* = 206)	(*n* = 246)	(*n* = 206)	(*n* = 193)	(*n* = 192)	(*n* = 225)	(*n* = 246)
Total iodide intake from dietary sources (food and drink) and supplements (µg/d)									
Geometric mean ^2,3^ (95% CI)	111 (100, 124)	127 (114, 141)	112 (99, 128)	136 (126, 146)	89 (79, 100)	76 (65, 88)	81 (71, 92)	98 (89, 107)	125 (118, 134)
Median (IQR)	129 (73, 196)	148 (76, 216)	134 (73, 218)	143 (94, 196)	105 (47, 164)	90 (41, 145)	87 (42, 149)	104 (64, 152)	132 (90, 173)
Iodide intake from dietary sources (food and drink) only (µg/d)									
Geometric mean ^2,3^ (95% CI)	74 (66, 82)	89 (79, 99)	86 (75, 98)	96 (89, 105)	84 (75, 94)	73 (63, 85)	78 (68, 88)	93 (85, 102)	101 (94, 108)
Median (IQR)	83 (46, 134)	89 (54, 152)	99 (57, 157)	101 (64, 142)	98 (45, 147)	86 (38, 144)	122 (41, 143)	100 (64, 149)	104 (69, 138)
Proportion of dietary iodide from each food source (%)									
Dairy	38%	41%	43%	42%	40%	34%	35%	38%	41%
Dairy with cereal	12%	9%	12%	11%	12%	12%	12%	12%	12%
Eggs	13%	11%	12%	12%	11%	12%	13%	13%	13%
Cereals	6%	6%	4%	5%	6%	6%	5%	5%	4%
White Fish	5%	7%	7%	9%	5%	7%	6%	10%	12%
Other seafood	2%	1%	2%	2%	1%	2%	1%	1%	2%
Poultry and red meat	11%	12%	10%	10%	15%	15%	17%	12%	9%
Fruit, vegetables and pulses	10%	8%	9%	7%	6%	10%	8%	6%	6%
Confectionery	2%	3%	1%	2%	3%	2%	1%	2%	2%
Other	1%	1%	1%	1%	1%	1%	<1%	1%	1%
Fish consumption (*n* (%)) ^4^									
Any white fish (*n* (%))	16 (7%)	22 (10%)	20 (10%)	50 (20%)	16 (8%)	16 (8%)	18 (9%)	42 (19%)	80 (33%)
Iodide intake from fish in fish consumers (µg/d)									
Geometric mean ^2,3^ (95% CI)	159 (85, 297)	178 (122, 262)	187 (127, 274)	67 (49, 90)	138 (87, 218)	302 (201, 454)	178 (114, 279)	81 (60, 110)	41 (32 to 51)
Supplement use (*n* (%))									
Used any supplement	220 (89%)	183 (84%)	159 (77%)	230 (94%)	63 (31%)	39 (20%)	26 (14%)	54 (24%)	233 (95%)
Used any supplement containing iodide	110 (45%)	98 (45%)	57 (28%)	160 (65%)	17 (8%)	8 (4%)	8 (4%)	23 (10%)	163 (66%)
Iodide intake from supplements (µg/d) geometric mean ^2,3^ (95% CI)									
In iodide supplement users ^5^	87 (77, 97)	89 (81, 98)	120 (111, 130)	54 (48, 60)	93 (63, 136)	118 (93, 150)	120 (80, 179)	52 (38, 72)	31 (28, 35)
Proportion of total iodide intake from supplements (%)									
In all participants	25%	23%	16%	25%	4%	2%	2%	4%	17%
In iodide supplement users ^5^	55%	51%	56%	38%	45%	55%	59%	34%	26%
Urinary iodine concentration (µg/L)									
Geometric mean (95% CI)	123 (111, 138)	107 (96,121)	127 (112, 143)	131 (121, 143)	92 (82, 104)	114 (103, 126)	128 (115, 143)	119 (109, 130)	133 (124, 142)
Median (IQR)	122 (77, 212)	122 (70, 176)	129 (75, 234)	135 (90, 207)	96 (56, 153)	114 (74, 171)	139 (83, 220)	121 (77, 190)	143 (90, 195)
Iodine/creatinine (µg/g)									
Geometric mean (95% CI)	123 (112, 134)	109 (101, 118)	137 (125, 151)	128 (119, 137)	97 (88, 108)	96 (87, 106)	107 (97, 118)	106 (98, 115)	120 (114, 128)
Median (IQR)	120 (75, 200)	104 (73, 150)	136 (91, 223)	126 (88, 187)	99 (61, 162)	94 (64, 147)	104 (71, 168)	111 (72, 160)	121 (90, 163)

^1^ Based on arithmetic mean for each individual, averaged over the number of visits completed within the time period. ^2^ To allow inclusion of any zero intakes, a constant of 1 was added before calculating geometric means, and subsequently subtracted from the derived mean and confidence intervals. ^3^ Geometric means for dietary sources of iodide do not sum to geometric mean for total dietary intake. ^4^ Among those consuming any fish during the time period. ^5^ Among those using any iodide-containing supplements during the time period.

**Table 3 nutrients-13-00230-t003:** Total, dietary and supplement iodide intake in all women and by ethnic background.

			All *n* = 246	Pakistani *n* = 131	White European *n* = 89	Other *n* = 26
Total iodide intake (µg/d)	All visits	Geometric mean (95% CI)	125 (118, 134)	117 (108, 127)	133 (121, 147)	143 (102, 199)
		Median (IQR)	132 (90, 173)	123 (81, 162)	138 (108, 182)	146 (98, 285)
	Antenatal visits	Geometric mean (95% CI)	136 (126, 146)	126 (114, 139)	150 (134, 168)	139 (101, 192)
		Median (IQR)	143 (94, 196)	129 (87, 190)	159 (114, 215)	142 (112, 226)
	Postnatal visits	Geometric mean (95% CI)	98 (89, 107)	89 (78, 100)	102 (89, 116)	151 (103, 222)
		Median (IQR)	104 (64, 152)	91 (60, 144)	110 (69, 151)	180 (81, 242)
Total iodide intake < UK RNI (140 µg/day)	Any visits	*n* (%)	145 (59%)	86 (66%)	46 (52%)	13 (50%)
	Antenatal visits	*n* (%)	119 (48%)	74 (56%)	33 (37%)	12 (46%)
	Postnatal visits	*n* (%)	160 (71%)	91 (75%)	59 (72%)	10 (48%)
Dietary iodide (µg/d)	All visits	Geometric mean (95% CI)	101 (94, 108)	96 (88, 105)	103 (94, 114)	119 (84, 169)
		Median (IQR)	104 (69, 138)	97 (66, 132)	111 (79, 140)	130 (72, 218)
	Antenatal visits	Geometric mean (95% CI)	96 (89, 105)	92 (82, 102)	103 (91, 117)	99 (69, 141)
		Median (IQR)	101 (64, 142)	91 (63, 132)	114 (69, 153)	91 (66, 173)
	Postnatal visits	Geometric mean (95% CI)	93 (85, 102)	86 (76, 97)	93 (82, 106)	151 (103, 222)
		Median (IQR)	100 (64, 149)	87 (59, 141)	104 (67, 143)	180 (81, 242)
Used any supplement containing iodide	Any visits	*n* (%)	163 (66%)	84 (64%)	64 (72%)	15 (58%)
	Antenatal visits	*n* (%)	160 (65%)	82 (63%)	63 (71%)	15 (58%)
	Postnatal visits	*n* (%)	23 (10%)	9 (7%)	14 (17%)	0 (0%)
Iodide from supplements in any users of supplements containing iodide ^1^ (µg/d)	All visits	Geometric mean (95% CI)	31 (28, 35)	29 (25, 34)	35 (28, 42)	29 (17, 48)
		Median (IQR)	32 (20, 52)	29 (18, 45)	35 (22, 62)	33 (21, 52)
	Antenatal visits	Geometric mean (95% CI)	54 (48, 60)	51 (44, 58)	58 (49, 69)	51 (31, 83)
		Median (IQR)	54 (36, 89)	50 (36, 75)	60 (36, 100)	62 (36, 96)
	Postnatal visits	Geometric mean (95% CI)	52 (38, 72)	46 (23, 93)	56 (38, 83)	/
		Median (IQR)	56 (36, 98)	56 (36, 88)	57 (40, 99)	/

^1^ Geometric mean and median iodide intake from supplements amongst those who had used any iodide-containing supplements at all during the specified time period.

**Table 4 nutrients-13-00230-t004:** Adjusted percent difference (95% CIs) in urinary iodine, thyroid hormone concentrations and goiter rate per 50 µg increase in iodide intake.

		UIC	I/Cr	TSH	Tg	fT3	fT4	fT3/fT4	Palpable Goiter ^1,2^
All visits	Total iodide intake	3% (1 to 5)	7% (5 to 9)	1% (−1 to 2)	−4% (−6 to −2)	−1% (−2 to 0)	0% (−1 to 1)	−1% (−1 to 0)	−5% (−18 to 8)
	Dietary iodide	2% (0 to 4)	5% (3 to 6)	1% (0 to 3)	−2% (−5 to 0)	0% (−1 to 1)	0% (−1 to 0)	0% (−1 to 1)	−6% (−19 to 8)
	Iodide from supplements	6% (1 to 11)	11% (7 to 15)	−3% (−6 to 0)	−12% (−17 to −7)	−5% (−7 to −4)	1% (−1 to 2)	−6% (−8 to −4)	−12% (−37 to 13)
Antenatal visits	Total iodide intake	4% (1 to 7)	5% (3 to 7)	1% (0 to 3)	−1% (−4 to 3)	0% (−1 to 1)	0% (−1 to 1)	0% (−1 to 1)	−21% (−32 to −9)
	Dietary iodide	3% (0 to 6)	4% (2 to 7)	2% (0 to 4)	−1% (−5 to 2)	0% (−1 to 1)	−1% (−2 to 0)	1% (0 to 2)	−31% (−44 to −18)
	Iodide from supplements	5% (−1 to 10)	8% (4 to 13)	−2% (−5 to 2)	4% (−3 to 10)	1% (−1 to 3)	2% (0 to 4)	−1% (−3 to 1)	12% (−20 to 44)
Postnatal visits	Total iodide intake	1% (−1 to 4)	3% (0 to 5)	−1% (−3 to 2)	−2% (−5 to 2)	1% (0 to 2)	0% (−1 to 2)	0% (−1 to 2)	/
	Dietary iodide	2% (−1 to 4)	3% (0 to 6)	0% (−3 to 3)	−2% (−6 to 2)	1% (0 to 2)	0% (−1 to 2)	0% (−1 to 2)	/
	Iodide from supplements	2% (−9 to 13)	4% (−5 to 13)	−12% (−23 to −2)	4% (−11 to 20)	0% (−4 to 4)	2% (−3 to 7)	−1% (−8 to 6)	/

Abbreviations: CI, confidence intervals; UIC, urinary iodine concentration; I/Cr, iodine/creatinine ratio; TSH, thyroid stimulating hormone; Tg, thyroglobulin; fT3, free triiodothyronine; fT4, free thyroxine. Adjusted for age, BMI, parity, ethnic background (Pakistani/White British or European/Other), NS-SEC, highest educational achievement, season. Smoking status and alcohol intake at the appropriate time-points were also adjusted for. Models using postnatal data additionally included adjustment for gestation length. ^1^ Results were not available for postnatal goiter because of small numbers with the binary outcome. ^2^ Grade 1 or 2 goiter, according to World Health Organization 1994 criteria. Estimate indicates % change in odds.

## Data Availability

This research was based on confidential information and records collected by the hospital, midwives and health visitors about the participants and their children during and after their pregnancy. The participants did not consent to access to this information by third parties for uses outside the scope of the project.

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
