# Peer review of "Prenatal and Postpartum Maternal Iodide Intake from Diet and Supplements, Urinary Iodine and Thyroid Hormone Concentrations in a Region of the United Kingdom with Mild-to-Moderate Iodine Deficiency"

_nutrients, 2021, doi:10.3390/nu13010230_

Round 1
Reviewer 1 Report
Threapleton and coworkers explored associations between daily iodine intake (estimated using 24-hour recalls) and urinary iodine concentrations (UIC), urinary iodine:creatinine ratio (I:Cr), TSH, thyroglobulin, free triiodothyronine, free thyroxine and goiter. The study sample included 246 pregnant women from a region of mild-to-moderate iodine insufficiency (Bradford, UK), with more than half of women with Pakistani heritage. Information on diet and supplement use, along with urine and blood samples, were obtained at around 12-, 26- and 36-weeks gestation, and 6, 18 and 30 weeks postpartum.
The geometric mean total iodide intake during pregnancy was 136 μg/day (95% CI 126-146), with a median of 143 μg/day (IQR 94-196). Based on estimates of iodine intake during pregnancy, almost 90% of the participants were found to be iodine insufficient according to WHO criteria for pregnant/lactating women (250 μg/day), and one half of them did not even meet the UK reference nutrient intake (RNI) of 140 μg/day set for all adults. Correlation of daily iodine intake with biochemical and clinical parameters of iodine status showed that lower iodine intake was associated with an increase in both thyroglobulin levels and goiter occurrence.
General comments
This study provides comprehensive estimates of daily iodine intake in a cohort of pregnant women from a mild-to-moderate iodine deficient area. The most important result is that pregnant women in UK are far from being iodine sufficient according to current WHO recommendations, which is a finding that deserves due attention because of the known detrimental effects of iodine deficiency on offspring. Unfortunately, the Authors do not provide similarly detailed information on maternal thyroid parameters, but only report estimates of the associations between iodine intake and the above measures as percent differences per 50 μg increase in total iodide intake. Though interesting, this kind of approach does not provide the reader with basic information on the actual relation between the thyroid status of this cohort of women and that given iodine intake.
Specific Comments
- Materials and Methods, section 1. Participants, page 3, lines 101-3. The Authors state that “Recruitment targeted women … with no medical or first-degree relative history of a thyroid condition”. How did the Authors ascertain the absence of thyroid diseases in their cohort of pregnant women? Did they perform thyroid function tests prior to enrollment? This is a relevant point, because many young women may experience mild thyroid conditions (mostly due to thyroid autoimmunity) that may go undetected until they are specifically investigated.
- Materials and Methods, section 1. Participants, page 3, line 112. Please, provide details on the analytes tested in blood samples at each visit. It is not clearly stated whether thyroid function parameters were obtained once in pregnancy/postpartum or serially during and after gestation. In either circumstance, mean (SD) and median gestational week of thyroid function testing should be indicated in the Results (see also comment #10)
- Materials and Methods, section 6. Statistical methods, page 4, lines 177-8. Vegan/vegetarian diets should be also included among known confounders of iodine intake.
- Materials and Methods, section 6. Statistical methods, page 4, line 180. What do the Author mean for “BMI at baseline”? Is it prior to conception or at enrollment? Please, specify.
- Results, section 1. Participant Characteristics, page 5, line 215. Categories of total iodine intake (lower, middle, upper third) should be reported in the text before the comparison of groups.
- Results, section 1. Participant Characteristics, table 1. The total number of women recruited is 246 but the sum of the women in each iodine category is 244. Please, check and correct.
- Results, section 2. Dietary intake, page 5, line 224. How was the distribution of iodine insufficiency (according to both WHO and UK RNI) in the three ethnic categories considered in this study? Was there any difference?
- Results, section 2. Dietary intake, page 5, lines 225-6. “Participants who did not use any iodide supplements were twice as likely to be below the UK RNI and three times as likely to be below the LRN”. Though raw data are provided in supplementary table 2, I believe it would be useful for the reader to know here the actual rate of iodine-supplemented women not meeting the WHO/UK RNI (and LRNI), along with the statistical significance of differences observed between groups.
- Results, section 2. Dietary intake, page 5, line 232. The finding that none of the women in this cohort used iodized salt is worthy of note and should be adequately emphasized in the discussion section. In this respect, please see and quote the following articles: Romano R, Jannini EA, Pepe M, Grimaldi A, Olivieri M, Spennati P, Cappa F, D’Armiento M. 1991 The effects of iodoprophylaxis on thyroid size during pregnancy. Am J Obstet Gynecol 164: 482–485; Moleti M, Lo Presti VP, Campolo MC, Mattina F, Galletti M, Mandolfino M, Violi MA, Giorgianni G, De Domenico D, Trimarchi F, Vermiglio F. 2008 Iodine prophylaxis using iodized salt and risk of maternal thyroid failure in conditions of mild iodine deficiency. J Clin Endocrinol Metab. 93:2616-21; Elizabeth N. Pearce, Iodine in Pregnancy: Is Salt Iodization Enough?, The Journal of Clinical Endocrinology & Metabolism, Volume 93, Issue 7, 1 July 2008, Pages 2466–2468
- Results, section5. Thyroid hormones, proteins and goiter. Analytical data on maternal thyroid parameters (TSH, FT3, FT4, Tg) at baseline and over the study period, along with information on the occurrence of maternal thyroid dysfunction (if any), should be provided.
- Results, section5. Thyroid hormones, proteins and goiter. Estimates of associations between iodide intake and UIC, I:Cr, TSH, fT3, fT4, fT3:fT4 ratio, and Tg are presented as percent changes per 50μg increase in total iodide intake. How was the rate of iodine intake increase chosen?
- Discussion, page 13, lines 323-6. Concerning goiter occurrence, the Authors should point out that their evaluation may not be so accurate, as they lacked an ultrasonographic estimation of thyroid size, which has been advocated as being more precise than palpation.
- Discussion, page 13, line 346. “…or results are consistent “. Correct or in our.
Reviewer 2 Report
The paper is well-written and the biomarkers of iodine nutrition is comprehensively studied.
There is a minor point about the evaluation of palpable goiter in pregnant women.
The thyroid gland enlarged by an average of 18% during pregnancy. The sensitivity and specificity of palpation is low in discriminating grade 0 from grade I goiter in pregnant women. Goiter surveys are usually done in school age children, and the median UIC, instead of goiter size, is recommended by WHO/ICCIDD/UNICEF for assessing iodine nutrition in pregnant women.
For the purpose of scientific research, goiter size could be studied but is preferably done by ultrasound.
Reviewer 3 Report
Please see attachment.
